# Analysis of Morphological Parameters and Body Composition in Adolescents with and without Intellectual Disability

**DOI:** 10.3390/ijerph20043019

**Published:** 2023-02-09

**Authors:** Bogdan Constantin Ungurean, Adrian Cojocariu, Beatrice Aurelia Abalașei, Lucian Popescu

**Affiliations:** 1Faculty of Physical Education and Sports, “Alexandru Ioan Cuza” University of Iași, 507184 Iași, Romania; 2Romania-Faculty of Physical Education and Sports, Doctoral School in Sports and Physical Education Science, “Alexandru Ioan Cuza” University of Iasi, 507184 Iași, Romania

**Keywords:** body mass composition, BMI, weight, intellectual disability

## Abstract

Compared to the tremendous volume of studies focusing on children and teenagers without disabilities, research regarding weight and body composition among young populations with an intellectual disability is relatively rare. Their number further decreases when we refer to specific age groups with intellectual deficits, such as children and adolescents younger than 18. In addition, studies are even scarcer when we wish to compare groups of subjects with different degrees of intellectual disability by gender. This study has a constative nature. The research sample comprises 212 subjects—girls and boys with an average age of 17.7 ± 0.2, divided into six groups by gender and type of intellectual disability. The parameters considered within the study include anthropometrical data and body composition determined using a professional device (Tanita MC 580 S). The findings of this study highlight the impact of intellectual disability on body composition in this age category. We hope it will help develop efficient strategies, recommendations, and intervention plans to ensure active participation in physical activities and categorisation within the optimal parameters of body composition indicators.

## 1. Introduction

The prevalence of obesity at the paediatric age has doubled in the past 30 years among pre-schoolers and adolescents, and it has tripled in the age group of 6–11 [1]. Child obesity is a predictive factor for morbidity and mortality among adults: up to 80% of obese children will be obese adults who will be highly susceptible to cancer, high blood pressure, strokes, hepatic and bile duct diseases, and osteoarthritis [2,3].

Youths with ID who are overweight or obese are also more likely to develop secondary conditions related to obesity, such as asthma, high blood cholesterol, diabetes, depression, and fatigue, compared to youths of the same population with normal weights [4]. Furthermore, children with intellectual disability (ID) can face several challenges concerning information processing (for instance, cognitive disorders, communication disorders, and limited mental function). Consequently, they have difficulties understanding and acquiring knowledge concerning health and developing healthy behaviours [5]. Recommended by the World Health Organisation, body mass index (BMI) and body composition are commonly used to measure obesity in different populations [6]. Considering these recommendations, it cannot be said whether these methods accurately measure body composition or fat distribution in populations with ID. Often, such individuals have specific anthropometry compared to those without disabilities [7]. Understanding the causes and effects of high body mass index or obesity remains essential when assessing the health states of persons with ID. For most populations with various forms of ID, BMI is a reasonable measure to identify the individuals most prone to the harmful effects of obesity. Current research [8] suggests that fat tissue is still the most detrimental to health and should be the target of any intervention measure.

Several methods have been implemented to measure body composition among persons with ID, such as waist circumference, skinfold measurements, as well as bioelectrical impedance analysis (BIA) [9]. However, the findings of a recent study have indicated that skinfold measurements provide too many errors among people with ID. For instance, Waninge et al. [10] have concluded that it is impossible to observe the measurement conditions, requiring the measurement of skinfolds three times precisely in the same spot on a person’s body. Previous research on body composition among people with ID focused solely on assessing body fat while alternating lean mass [11]. However, total muscle mass, of which the skeletal mass is a primary component related primarily to the physical function of an individual, is considered, for the most part, the most significant compartment of the body in assessing the physiological and nutritional state. In addition, a recent study on children with intellectual disabilities has reported that bioelectrical impedance analysis (BIA) is more viable than skinfold measurements [12,13]. Because communication with people with ID and particularly with a severe intellectual disability is challenging and no other non-invasive body composition measuring tools are available, the feasibility and viability of BIA measurements can be more relevant.

This study aimed to assess a series of morphological and body compositions among children with and without intellectual disability to characterise the morphofunctional normality and its disturbance. The data obtained after using the statistical–mathematical indicators will be analysed in relation to the literature. The research tasks may be summarised as collecting and studying the literature and processing the data collected based on the statistical–mathematical methods to provide an objective interpretation and to elaborate the conclusions of the research conducted.

Primary assumption 1: 

**Hypothesis** **1.**
*Intellectual disability influences the morphological parameters among children with intellectual disability.*


Secondary assumptions:

**Hypothesis** **1.1.***The type of disability influences some parameters of body composition*.

**Hypothesis** **1.2.**
*There are interaction effects of the intellectual disability and gender variables with some morphological parameters.*


## 2. Materials and Methods

Ethics: All the procedures in this study conformed with the 1964 Declaration of Helsinki and its subsequent amendments. We conducted the research with the approval of the Ethics Commission of scientific research No. 10/2020, and the date of approval was 7 October 2020.

The research per se began a long time ago through discussions with the Physical Education and Sports teachers within the centres schooling children with intellectual disability, meetings with specialists in the field, and collecting and studying the literature. The measurements began in April 2021 and continued until November 2022, considering the pandemic context of that moment.

Participants. The activities took place in the gymnasiums of the academic units, as well as in the physical therapy practices of the “Sf. Andrei” School Centre Gura Humorului, Suceava County; the “Constantin Păunescu” School Centre Iaşi; “Elisabeta Polihroniade” Inclusive Education School Centre Vaslui; “Emil Gârleanu” Special School No. 1 Galați. It is worth mentioning that the measurements were carried out in the first part of the day (in 9–13) for all the groups. These institutions educate children with different types of intellectual disabilities. The inclusion criterion in the study was that ID should not be associated with other disabilities. The subjects’ parents or tutors signed a protocol at the beginning of the school year. This study included 212 subjects of the aforementioned educational establishments, distributed into six groups by gender and type of disability, as illustrated in Table 1.

Procedure. Morphological parameters and some components of body composition represent the dependent variables.

Height—To accurately measure a subject’s height, they must stand barefoot, with their back, head, and heels touching a vertical wall and their head facing forward. Using a telemeter, the distance from the ground to the perpendicular wall projection of the vertex point (the highest cranial point), determined using a 90°-angle object (e.g., a set square) placed with one of the sides on the vertex and one on the wall, is measured. It is recorded in centimetres and subdivisions of 0.5 cm. For the measurements made on this group of subjects, we used a telemeter with a Bosch GLM 80 laser to obtain an accurate measurement.We used a professional device called TANITA MC 580 S and dedicated analysis software to determine body composition. BIA (bioelectrical impedance analysis) is a technique used to measure body composition. The technology for analysing bioelectrical impedance involves the passage of a low-intensity electrical current (around 500 µA) from the electrodes under the soles to those held in the hands. Professional models provide a segmental analysis; the seven electrodes offer additional information for each foot, arm, and area (abdominal). The electrical signal passes rapidly through the water in the hydrated muscle tissue, but it faces resistance from the adipose tissue. This resistance, also known as impedance, is measured and introduced into scientifically validated Tanita equations to calculate body composition measurements. TANITA multiple-frequency monitors can measure the bioelectrical impedance analysis on three or six different frequencies. Additional frequencies provide an exceptional precision level compared to monitors with one or two frequencies. Lower frequencies measure the impedance outside the cell membrane. Higher frequencies can penetrate the cell membrane, measuring the impedance at the lower and higher levels, thus possibly estimating extracellular and intracellular water and total body water. Such information is essential to provide data on a person’s health and indicate potential health risks.

TANITA PRO SOFTWARE Version 3.4.5—The Tanita PRO software pack was developed in partnership with an essential medical software developer (Medizin & Sevice GmbH, Chemnitz, Germany). The software can store and analyse the data from the Tanita MC 580 S monitor. In conformity with the EU Regulations, the software is medically approved and observes the standing Regulations [14] Eur lex. (The Medical Devices Directives, Directive 93/42/EEC of the Council of 14 June 1993 on medical devices). The use of TANITA MC580 S and TANITA PRO SOFTWARE generates many measurements; the most representative introduced within this study as dependent variables are body mass (kg), body mass index (BMI kg/h^2^), body fat (%), muscle mass (%), basal metabolic rate (kcal), body fat (Kg), muscle mass (Kg), and skeletal muscle mass (SMM).

Statistical Analysis. MANOVA (two-way MANOVA)—The two-way multivariate analysis of variance (two-way MANOVA) is often considered an extension of the two-way ANOVA for situations in which there are two dependent variables. The primary purpose of the two-way MANOVA is to understand if there is an interaction between two independent variables on the other dependent variables combined. Considering the significant number of data (over 200 subjects), it is recommended to use a skewness statistical indicator to test the normality of data distribution, which evaluates the asymmetry degree of distribution and the kurtosis indicator. SPSS provides both tests. In [15], two z thresholds are proposed by the number of subjects tested. For a more significant number of data (over 150–200), the z threshold is 1.96 [16]. The Kruskal–Wallis H test is a rank-based non-parameter test that can determine if there are statistically significant differences between two or more groups of an independent variable on a continuous or ordinal dependent variable. The Kruskal–Wallis H test may be used when the data do not observe the unidirectional ANOVA assumptions. It occurs if (a) the data are not normally distributed or (b) there is an ordinal dependent variable. Descriptive analyses—in SPSS 20.0, through graphical and numerical synthesis, leaving some of the information out to gain relevance; The Tukey Procedure—(honestly significant difference—HSD) is a method based on q statistics, and it is preferred for group comparisons, two by two. The technique is effective for multiple group comparisons when groups are uneven.

## 3. Results

To determine the tests used for data interpretation, we considered the values of skewness and kurtosis indicators concerning data distribution.

After analysing the values of kurtosis (Table 2), we note that in four dependent variables (body mass—Kg, BMI (kg/m^2^), muscle mass %, and body fat Kg), for the nine dependent variables, there is no normal data distribution, which made us use a nonparametric test (Kruskal–Wallis H). For the other five dependent variables (height—cm, body fat %, BMR—kcal, muscle mass—Kg, and SMM) with normal data distribution, we used the Manova test. In this respect, we wished to determine the existence of a statistically significant interaction effect by interpreting the multivariate testing.

It tests the null assumption according to which the covariance matrices of the dependent variables are equal between groups [17]. Consequently, we relied on Pillai’s Trace (*p* = 0.008); as shown in Table 3, there is a statistically significant interaction effect between gender and type of disability on the dependent variables combined (*p* < 0.05).

In this situation, the assumption of the homogeneity of covariance matrices was not observed (Table 4), as assessed using Box’s M test (*p* < 0.001). Box’s M test is known to be very sensitive when multivariate normality is not observed, leading to a statistically significant result due to non-normality [18]. However, the MANOVA test is considered robust to the non-observance of this assumption. Hence, if the assumption of covariance equality is not observed, we may continue, regardless of whether the groups have similar sizes. Though Wilks’ Lambda test is usually recommended, Pillai’s Trace test is more robust, and it is a reliable choice when the samples are uneven, and the M matrix is present (Table 4); Box’s test—significant covariance equality (*p* < 0.001).

As illustrated in Table 5, with the multiple comparisons by gender, several significant differences were recorded between the dependent variables with normal distribution for the groups of boys. The only dependent variable not influenced by the type of intellectual disability was body fat %. It is worth noting that we found significant differences (*p* < 0.05) in the other four dependent variables (height, BMR kcal, SMM, and muscle mass—kg) between the group of boys without intellectual disability and the group of boys with moderate intellectual disability (height *p* = 0.013, BMR kcal *p* = 0.005, SMM *p* = 0.001, muscle mass—kg *p* < 0.001) and between the group of boys without intellectual disability and the group of boys with severe intellectual disability (height *p* < 0.001, BMR kcal *p* = 0.009, SMM *p* = 0.001, muscle mass—kg *p* < 0.001). However, between the group of boys with moderate intellectual disability and the group of boys with severe intellectual disability, we found no significant differences in any dependent variable.

We performed a Kruskal–Wallis H test to determine the existence of significant differences concerning the dependent variables without a normal distribution between the three groups of boys. The median body mass (kg) scores were statistically significantly different between groups, *p* = 0.005. The median scores for IMC, body fat (kg), and muscle mass (%) were not statistically significantly different between groups *p* ˃ 0.05. The paired comparisons for body mass (kg) were performed using Dunn’s procedure, with a Bonferroni correction for multiple comparisons. Here, we feature the adjusted values of p. The post hoc analysis revealed significant differences in the median scores of body mass (kg) between the groups of boys with severe intellectual disability and without intellectual disability (*p* = 0.024) and between the groups of boys with moderate intellectual disability and without intellectual disability (*p* = 0.014). Between the groups of boys with severe intellectual disability and moderate intellectual disability, we found no significant differences (Appendix A).

Concerning the groups of girls, as shown in Table 6, we only recorded significant differences in the dependent variable of height between the group of girls without intellectual disability and the group of girls with moderate intellectual disability (*p* = 0.036) and between the group of girls without intellectual disability and the group of girls with severe intellectual disability (*p* = 0.024). Regarding the other four dependent variables (BMR kcal, SMM, body fat %, muscle mass—kg), we found no significant differences between the three groups of girls.

To analyse the data without a normal distribution in the groups of girls, we applied the Kruskal–Wallis H test. We found no significant differences between the four groups for any dependent variable (*p* ˃ 0.05).

After analysing the data by gender, we found significant differences for the dependent variables with a normal distribution (Table 7). We identified values of *p* < 0.05 between almost all pairs of subjects, except for the SMM variable, between the group of girls with severe intellectual disability and the group of boys with severe intellectual disability (*p* = 0.137).

For the dependent variables without a normal distribution, we recorded significant differences in three of the four dependent variables. The only dependent variable without differences by gender was body mass index (for BMI *p* ˃ 0.05) (Table 8).

## 4. Discussion

In this study, significant differences (*p* < 0.05) were found in the groups of boys for five of the nine dependent variables, particularly between the group of boys without intellectual disability and the group of boys with moderate intellectual disability, as well as between the group of boys without intellectual disability and the group of boys with severe intellectual disability. However, between the groups of boys with different types of intellectual disability, no significant differences could be reported. One of the dependent variables not influenced by the type of disability or gender was body mass index. Though no statistical differences between groups were recorded, the means per group (Table 8) of the BMI exceeded the WHO guidelines [19]. Recent research shows similar findings among children without intellectual disability [20,21] and with intellectual disability [22]. The combined prevalence of overweight and obesity among European teenagers is 22–25% [23]. This figure has increased constantly in the past few decades, and now it appears to be rising faster in Western countries.

Nonetheless, studies show differences in socioeconomic status and geographical position [24,25]; an increase was noted in children and adolescents with financial difficulties and those with intellectual disability [26]. Because they go through a stage of growth and development, body composition modifications are to be expected at this age. In boys, muscle mass indicators will increase, thus recording statistically significant differences by gender for all three categories (*p* < 0.05, Table 8), as sexual hormones lead to a substantial increase in muscle mass. Among girls, on the other hand, puberty development involves a period of fat tissue storage [27]. As illustrated in Table 8, the average values of the dependent variable (body fat—kg) were higher than the boys’ groups in all three categories; we found significant differences for *p* < 0.05. This aspect is seen as a physiological preparation for birth, where extra energy is necessary to have and feed the new-born [28]. The differences can be due to multiple features of persons with ID, which may be approached individually or collectively, ascribing them to idleness, and social barriers hindering access to exercise programs, which affect the BMI and BMC alike [29]. Substantial body composition modifications among teenagers are due to puberty, the development process inherent to this stage of life, with significant differences between genders. Thus, it is challenging in longitudinal studies to distinguish between unhealthy weight gains and natural body composition modifications [30].

Unfortunately, few studies focus on the relationships between body composition in populations with different types of intellectual disability. It is even more challenging to find recent studies featuring differences in this population by gender. Against this backdrop, this study aimed to assess the relationship between body composition (assessed using a Tanita 580 S professional device relying on BIA technology) in teenagers with and without intellectual disability by gender. The substantial differences between genders concerning the body composition of adolescents argue for the presentation of our study findings by gender.

This study has several limitations: the data analysis did not assess how much the subjects exercised and how many calories they burnt; the body composition using BIA technology was assessed in this research, which is both commonly used and reliable, but there was no comparison with other methods. This limitation can be considered as a future research topic to include diet and physical activity measurement. However, the study has several strong points, too: it is among the few pieces of research discussing body composition and body mass index by different types of intellectual disability and gender.

## 5. Conclusions

Bioelectrical impedance analysis (BIA) is a commonly used technology in research concerning body composition because it is non-invasive and quick, and the data are highly reliable. It can be moved to various locations and is particularly easy to use for populations with different types of intellectual disability. This research has confirmed that the primary factors of body composition (body mass (Kg), body mass index (BMI kg/h^2^), body fat (%), muscle mass (%), basal metabolic rate (BMR kcal), body fat (Kg), muscle mass (Kg), skeletal muscle mass (SMM), and the morphological indicators (height and weight) may be influenced by both the type of disability and gender. The prevalence of overweight and obesity among people with intellectual disabilities was similar between male and female subjects. This shows an increasing trend with age. Body composition is an essential determinant of health states and nutritional indicators. Hence, body composition analysis is crucial in assessing such populations’ physiological and pathological states. Body composition evaluation has become increasingly popular in clinical practice, primarily due to the constant increase in the obesity rate. The results obtained in this study may help to develop intervention strategies for the treatment of obesity, provided that decision-makers prioritise the treatment of people with intellectual disabilities.

## Figures and Tables

**Table 1 ijerph-20-03019-t001:** Distribution of the subjects by age, cases, and educational establishments.

Subjects	Gen	N	Age (Mean ± Std. Dev.)	Case Observation
Group 1 (WID) Without intellectual disability	M	44	17.7 ± 0.9	BWID
Group 2 (WID) Without intellectual disability	F	55	17.2 ± 0.7	GWID
Group 3 (MID) Moderate intellectual disability	M	57	17.05 ± 0.7	BMID
Group 4 (MID) Moderate intellectual disability	F	22	16.6 ± 0.8	GMID
Group 5 Severe intellectual disability (SID)	M	23	17.4 ± 0.8	BSID
Group 6 Severe intellectual disability (SID)	F	11	17.1 ± 0.8	GSID

**Table 2 ijerph-20-03019-t002:** Synthetic table featuring the skewness and kurtosis values for the morphofunctional parameters.

	Height cm	Body Mass kg	BMI (kg/m^2^)	Body Fat %	Muscle Mass %	BMR (kcal)	Body Fat Kg	Muscle Mass Kg	SMM
Valid N	212	212	212	212	212	212	212	212	212
Missing	0	0	0	0	0	0	0	0	0
Mean	167.416	62.413	22.128	20.759	44.554	1612.23	13.559	46.529	27.735
Std. Deviation	9.5748	14.6358	4.4270	8.5665	5.5875	283.509	8.2494	9.5670	5.9110
Skewness	0.295	1.216	1.446	0.560	−1.155	0.736	1.859	0.737	0.843
Std. Error of Skewness	0.167	0.167	0.167	0.167	0.167	0.167	0.167	0.167	0.167
Kurtosis	−0.272	**2.025**	**3.213**	0.325	**2.659**	0.528	**5.477**	0.785	1.012
Std. Error of Kurtosis	0.333	0.333	0.333	0.333	0.333	0.333	0.333	0.333	0.333
Minimum	145.0	33.1	13.7	4.6	19.6	1042	3.0	25.3	15.1
Maximum	193.0	121.1	41.7	51.1	54.0	2726	56.4	83.2	49.5

BMI—body mass index, BMR—basal metabolic rate, SMM—skeletal muscle mass.

**Table 3 ijerph-20-03019-t003:** Table featuring the multivariate testing.

Effect		Value	F	Hypothesis df	Error df	Sig.
Intercept	Pillai’s Trace	0.998	11,345.775 ^b^	10.000	197.000	0.000
	Wilks’ Lambda	0.002	11,345.775 ^b^	10.000	197.000	0.000
	Hotelling’s Trace	575.928	11,345.775 ^b^	10.000	197.000	0.000
	Roy’s Largest Root	575.928	11,345.775 ^b^	10.000	197.000	0.000
Disability type	Pillai’s Trace	0.662	9.792	20.000	396.000	0.000
	Wilks’ Lambda	0.407	11.169 ^b^	20.000	394.000	0.000
	Hotelling’s Trace	1.286	12.600	20.000	392.000	0.000
	Roy’s Largest Root	1.136	22.500 ^c^	10.000	198.000	0.000
Gender	Pillai’s Trace	0.582	27.391 ^b^	10.000	197.000	0.000
	Wilks’ Lambda	0.418	27.391 ^b^	10.000	197.000	0.000
	Hotelling’s Trace	1.390	27.391 ^b^	10.000	197.000	0.000
	Roy’s Largest Root	1.390	27.391 ^b^	10.000	197.000	0.000
Disability type * Gender	Pillai’s Trace	0.181	1.975	20.000	396.000	* **0.008** *
	Wilks’ Lambda	0.825	1.989 ^b^	20.000	394.000	* **0.007** *
	Hotelling’s Trace	0.204	2.003	20.000	392.000	* **0.007** *
	Roy’s Largest Root	0.154	3.057 ^c^	10.000	198.000	* **0.001** *

Design: Intercept + type + gender + type of disability * gender; ^b^. Exact statistic; ^c^. The statistic is an upper bound on F that yields a lower bound on the significance level.

**Table 4 ijerph-20-03019-t004:** Box’s Test—covariance matrix.

Box’s M	866.442
F	10.459
df1	75
df2	11470.924
Sig.	** *0.000* **

Design: Intercept + type + gender + type of disability * gender. F—approximation used to compute the significance; df1, df2—degrees of freedom for the Fisher distribution; Sig.—significance threshold for α < 0.001.

**Table 5 ijerph-20-03019-t005:** Multiple comparisons for the dependent variables with normal distribution in the groups of boys.

Dependent Variable	Height	BMR Kcal	SMM	Body Fat %	Muscle Mass—kg
Group	M ± Std. Dev.	M ± Std. Dev.	M ± Std. Dev.	M ± Std. Dev.	M ± Std. Dev.
	Sign. Thres.	Sign. Thres.	Sign. Thres.	Sign. Thres.	Sign. Thres.
BWID	176.38 ± 8.86	1865.4 ± 280.02	34.32 ± 6.03	15.8 ± 6.49	56.8 ± 9.3
BMID	168.59 ± 7.89	1717.2 ± 228.5	28.75 ± 4.3	18.96 ± 8.88	48.18 ± 7.26
	** *p = 0.013* ** *****	** *p = 0.009* ** *****	** *p = 0.001* ** *****	*p* = 0.124	** *p = 0.001* ** *****
BWID	176.38 ± 8.86	1865.4 ± 280.02	34.32 ± 6.03	15.8 ± 6.49	56.8 ± 9.3
BSID	170.1 ± 9.04	1664.3 ± 206.9	27.62 ± 4.77	17.37 ± 8.19	47.75 ± 7.85
	** *p = 0.013* ** *****	** *p = 0.005* ** *****	** *p = 0.001* ** *****	*p* = 0.726	** *p = 0.001* ** *****
BMID	168.59 ± 7.89	1717.2 ± 228.5	28.75 ± 4.3	18.96 ± 8.88	48.18 ± 7.26
BSID	170.1 ± 9.04	1664.3 ± 206.9	27.62 ± 4.77	17.37 ± 8.19	47.75 ± 7.85
	*p* = 0.745	*p* = 0.657	*p* = 0.638	*p* = 0.701	*p* = 0.975

* Significance threshold for ***p* < *0.05.***

**Table 6 ijerph-20-03019-t006:** Multiple comparisons for the dependent variables with normal distribution in the groups of girls.

Dependent Variable	Height	BMR Kcal	SMM	Body Fat %	Muscle Mass—kg
Group	M ± Std. Dev.	M ± Std. Dev.	M ± Std. Dev.	M ± Std. Dev.	M ± Std. Dev.
	Sign. Thres.	Sign. Thres.	Sign. Thres.	Sign. Thres.	Sign. Thres.
GWID	163.16 ± 6.49	1395 ± 160.7	24.12 ± 3.09	23.98 ± 5.78	40.45 ± 5.19
GMID	159.17 ± 6.28	1412.4 ± 151.7	23.12 ± 3.19	27.8 ± 8.4	39.08 ± 5.48
	** *p = 0.036* ** *****	*p* = 0.911	*p* = 0.460	*p* = 0.084	*p* = 0.595
GWID	163.16 ± 6.49	1395 ± 160.7	24.12 ± 3.09	23.98 ± 5.78	40.45 ± 5.19
GSID	157.61 ± 4.86	1431.9 ± 223.7	23.6 ± 4.45	26.7 ± 9.33	39.52 ± 7.46
	** *p = 0.024* ** *****	*p* = 0.783	*p* = 0.885	*p* = 0.455	*p* = 0.870
GMID	159.17 ± 6.28	1412.4 ± 151.7	23.12 ± 3.19	27.8 ± 8.4	39.08 ± 5.48
GSID	157.61 ± 4.86	1431.9 ± 223.7	23.6 ± 4.45	26.7 ± 9.33	39.52 ± 7.46
	*p* = 0.780	*p* = 0.947	*p* = 0.918	*p* = 0.915	*p* = 0.975

* Significance threshold for ***p* < *0.05***.

**Table 7 ijerph-20-03019-t007:** Multiple comparisons for the dependent variables with normal distribution by gender (morphological parameters).

Dependent Variable	Height	BMR kcal	SMM	Body Fat %	Muscle Mass—kg
Group	M ± Std. Dev.	M ± Std. Dev.	M ± Std. Dev.	M ± Std. Dev.	M ± Std. Dev.
	Sign. Thres.	Sign. Thres.	Sign. Thres.	Sign. Thres.	Sign. Thres.
BWID	176.38 ± 8.86	1865.4 ± 280.02	34.32 ± 6.03	15.8 ± 6.49	56.8 ± 9.3
GWID	163.16 ± 6.49	1395 ± 160.7	24.12 ± 3.09	23.98 ± 5.78	40.45 ± 5.19
	** *p < 0.001* ** *****	** *p < 0.001* ** *****	** *p < 0.001* ** *****	** *p < 0.001* ** *****	** *p < 0.001* ** *****
BMID	168.59 ± 7.89	1717.2 ± 228.5	28.75 ± 4.3	18.96 ± 8.88	48.18 ± 7.26
GMID	159.17 ± 6.28	1412.4 ± 151.7	23.12 ± 3.19	27.8 ± 8.4	39.08 ± 5.48
	** *p < 0.001* ** *****	** *p < 0.001* ** *****	** *p < 0.001* ** *****	** *p < 0.001* ** *****	** *p < 0.001* ** *****
BSID	170.1 ± 9.04	1664.3 ± 206.9	27.62 ± 4.77	17.37 ± 8.19	47.75 ± 7.85
GSID	157.61 ± 4.86	1431.9 ± 223.7	23.6 ± 4.45	26.7 ± 9.33	39.52 ± 7.46
	** *p < 0.001* ** *****	** *p = 0.043* ** *****	*p* = 0.137	** *p = 0.011* ** *****	** *p = 0.025* ** *****

* Significance threshold for ***p* < *0.05***.

**Table 8 ijerph-20-03019-t008:** Multiple comparisons for the dependent variables without normal distribution by gender.

Dependent Variable	Body Mass—Kg	BMI (kg/m^2^)	Muscle Mass %.	Body Fat Kg
Group	M ± Std. Dev.	M ± Std. Dev.	M ± Std. Dev.	M ± Std. Dev.
	Sign. Thres.	Sign. Thres.	Sign. Thres.	Sign. Thres.
Group 1	71.76 ± 15.1	22.95 ± 3.91	47.6 ± 4.12	11.95 ± 6.9
Group 2	56.65 ± 10.64	21.26 ± 3.61	43.02 ± 3.27	14.45 ± 6.17
	** *p = 0.021* ** *****	*p = 0.207*	** *p < 0.001* ** *****	** *p = 0.021* ** *****
Group 3	63.61 ± 14.96	22.43 ± 5.08	45.11 ± 6.66	12.92 ± 9.26
Group 4	58.06 ± 14.68	22.33 ± 4.86	40 ± 6.18	17.13 ± 10.11
	** *p = 0.005* ** *****	*p = 0.207*	** *p < 0.001* ** *****	** *p = 0.005* ** *****
Group 5	60.88 ± 10.68	20.99 ± 3.44	46.8 ± 4.63	10.99 ± 6.75
Group 6	59.1 ± 18.31	23.4 ± 6.6	41.45 ± 5.29	17 ± 12.45
	*p* = 0.079	*p = 0.207*	** *p = 0.003* ** *****	*p* = 0.079

* Significance threshold for ***p* < *0.05***.

## Data Availability

All relevant data are within the study, and raw data are available on request.

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
