# Peer review of "Analysis of Morphological Parameters and Body Composition in Adolescents with and without Intellectual Disability"

_ijerph, 2023, doi:10.3390/ijerph20043019_

Round 1

Reviewer 1 Report

Introduction:

Line 30-32 the text refers to literature, but it is not cited in either of these two lines.

Line 39, restructure the way you list the focus of the different studies, for example by adding "or Down Syndrome".

Line 39-42, what literature substantiates the low levels of physical activity among youth with intellectual disabilities? Also, the next sentence provides the same information as these lines.

I miss the description of a study objective.

Material and Methods:

Please divide the Material and Method section into subsections: "Participants", "Procedure", "Statistical Analysis".

Both the objective and hypotheses should appear in the Introduction section.

Following the structure of Table 1, I recommend that you make sure that the observation case in the last column is correct, SID instead of WMD.

The morphological parameters should be better structured to facilitate the reader's understanding, use subsections or bulleted lists.

 Please try not to write in the first-person plural.

Restructure this section completely, using clear and easy-to-follow language. Also include the information in different subsections.

Results:

The first part of the Results should be the "Statistical Analysis" subsection in Material and Method.

Please clarify in the table footnote (Table 2) the abbreviations included in the table.

Please specify in the text that you are analyzing each of the tables.

Again, please clarify the abbreviations in Table 4.

You should reorganize Table 5 so that the differences are organized by rows. For example, comparisons between groups 1 and 3 in one row. In addition, and to my knowledge, the differences shown in the text between groups do not resemble those described in the table. Group 1 and 3 in the text = (Height p=0.013, BMR kcal p=0.005, SMM p=0.001, Muscle mass - kg 177 p<0.001) but in the table (Height p<0.001, BMR kcal p=0.009, SMM p=0.001, Muscle mass - kg 177 p<0.001). Review these questions.

Figure 1, please construct new tables. Do not insert SPSS screenshots.

Again, Table 6 shows the same deficiencies as Table 5. If you are going to compare differences between Group 2 and 6, do not isolate Group 6 in the first column.

 Again, Figure 2 is a screen shot.

Tables 7 and 8 are correct in showing the differences between groups, please take it as an example for the previous ones. Good work.

 Include in all table captions the abbreviations that the tables contain.

Discussion:

Lines 242 to 246, they cannot assert changes in these ages if they do not cite literature.

The same occurs from 252 to 256.

I miss a subsection on future lines of research, also the practical implications are not entirely clear.

Conclusions:

Correct, no comments.

Reviewer 2 Report

EVALUATION

Thank you for sending this important research for review. Unequivocally, this research has tremendous benefits in indicating the influence of disability type and gender on body weight and body composition.

Introduction

In reading the introduction, I noticed several shortcomings that can be improved by careful reading and consistency in writing. For example, the authors talked a lot about PA and its benefits for people with and without intellectual disabilities, however, everything else in the paper talks about body type and disability. In other words, the introduction that the authors made is much more valid for a paper that studies the effects of PA on weight and body composition more than a paper that deals with the relationship between disability and body constituents. I suggest that the authors revise the Introduction to clarify the scope of the study and the reasons for this choice.

 Materials and Methods

 Lines 61-68 and 70-77: please move it to the end of the introduction.

Please provide a brief description of the participants and how they were recruited for this study.

Did participants sign a consent form?

The protocols must be described concisely (principle, procedure, materials used with their manufacture and origin, variable(s)).

 Results

Lines 134-145: please move them to the end of the "Materials and Methods" section in a separate section: "Statistical Procedures".

Please be brief and clear and avoid unnecessary details and explanations. You are not expected to explain everything.

Line 132: Using only the statistical indicator of skewness to test the normality of the data distribution requires a reference.

Figures are useless. They do not add anything to the study.

Please replace "groups 1, 2 ......" with abbreviations that reflect the characteristics of these groups such as BWID for boys with intellectual disability.

I think that Table 5 is incorrect.

 Discussion

The discussion is too short (50 lines including 13 for the summary of key results).

This part of the manuscript deserves special attention from the authors. They should explain their key findings and the mechanisms supporting those findings, primarily those related to disability, as some findings are somewhat surprising. Again, the authors limited themselves to suggesting physical activity as a determining factor, while many other factors play a role, including diet, which the authors completely ignored. If any, these factors may be cited as study limitations.

 Conclusions

 Authors are encouraged to briefly review the main arguments and recall the most important evidence in this section. This section should be revised.
